# Effect of *Sechium edule* var. *nigrum spinosum* (Chayote) on Oxidative Stress and Pro-Inflammatory Markers in Older Adults with Metabolic Syndrome: An Exploratory Study

**DOI:** 10.3390/antiox8050146

**Published:** 2019-05-27

**Authors:** Juana Rosado-Pérez, Itzen Aguiñiga-Sánchez, Edelmiro Santiago-Osorio, Víctor Manuel Mendoza-Núñez

**Affiliations:** 1Research Unit on Gerontology, FES Zaragoza, National Autonomous University of Mexico, Mexico City 09230, Mexico; juanaropez@yahoo.com.mx; 2Hematopoiesis and Leukemia Laboratory, Research Unit on Cell Differentiation and Cancer, FES Zaragoza, National Autonomous University of Mexico, Mexico City 09230, Mexico; liberitzen@yahoo.com.mx (I.A.-S.); edelmiro@unam.mx (E.S.-O.)

**Keywords:** *Sechium edule*, chayote, metabolic syndrome, oxidative stress, inflammatory markers, older adults

## Abstract

Metabolic syndrome (MetS) is a risk factor for cognitive deterioration and frailty in older adults. In this regard it has been shown that oxidative stress (OxS) and chronic inflammation are involved in the pathophysiology of these alterations. Harmless antioxidant and anti-inflammatory therapeutic alternatives have been proposed, such as the consumption of *Sechium edule* (chayote), but the evidence is inconclusive. For this reason, an exploratory study of a single group chosen by convenience sampling, including 12 older adults, with an average age of 71 ± 6 years (10 women and 2 men) with a diagnosis of MetS according to the National Cholesterol Education Program Adult Treatment Panel III (NCEP/ATP III) criteria. This exploratory study aimed to determine the effect of the consumption of the dried fruit powder supplement of *Sechium edule* var. *nigrum spinosum* (500 mg, 3 times per day) for six weeks on the markers of OxS in elderly adults with MetS. All participants’ OxS markers were measured before and after treatment. There was a statistically significant decrease in the concentration of lipoperoxides (baseline, 0.289 ± 0.04 vs. post-treatment, 0.234 ± 0.06 μmol/L, *p* < 0.05), together with a significant increase in total antioxidant status (baseline, 0.97 ± 0.18 vs. post-treatment, 1.2 ± 0.12 mmol/L, *p* < 0.05). In this sense, the oxidative stress index showed a statistically significant decrease (baseline, 1.7 ± 0.78 vs. post-treatment, 0.75 ± 0.87, *p* < 0.05). A statistically significant decrease in the concentration of TNF-α after treatment was also found (baseline, 5.3 ± 1.4 vs. post-treatment, 3.5 ± 1.3, *p* < 0.05).Our findings suggest that the consumption of the dry fruit of *Sechium edule* has an antioxidant and anti-inflammatory effect in older adults with metabolic syndrome.

## 1. Introduction

Metabolic syndrome (MetS) is a group of biochemical and clinical alterations characterized by insulin resistance, dyslipidemia, inflammation, coagulation disorders, hypertension and obesity [1,2]. The prevalence of MetS in older adults is more than 50%, and has been shown to be a risk factor for cognitive deterioration and frailty, as well as for type 2 diabetes mellitus and cardiovascular diseases [3,4,5,6].

It has been shown that oxidative stress (OxS) and chronic inflammation (CI) are involved in the pathophysiology of these alterations. For this reason, complementary antioxidant and anti-inflammatory therapeutic alternatives have been proposed [7,8,9,10,11]. In this context it has been reported that older adults often consume herbal medicinal products, such as the *Sechium edule* (chayote) [12,13].

The chayote is an edible plant of the Cucurbitaceae family with a high nutrient content. The reported nutrients include aspartic acid, glutamic acid, alanine, proline, serine, tyrosine, threonine and valine; vitamins such as thiamine, riboflavin, niacin, vitamin A and ascorbic acid; and the minerals calcium, phosphorus, iron, nitrogen, copper, zinc, manganese and potassium [13,14,15].

Phytochemical studies have revealed the presence of sterols, non-phenolic alkaloids, triterpenes and saponins, as well as flavonoids, in both fruits and seeds [15,16,17]. This explains the wide use of *Sechium edule (S. edule)* with curative purposes in México and worldwide. Likewise, in some studies it has been reported to have antioxidant, anti-inflammatory, hypoglycemic, hypotensive as well as lipogenesis inhibition properties [18,19]. It could therefore be an alternative treatment for the control of MetS. For this reason, the aim of this study was to determine the effect of the consumption of the dried fruit powder supplement derived from *S. edule* on both CI and OxS markers in older adults with MetS.

## 2. Materials and Methods

### 2.1. Design and Subjects

An exploratory pre-experimental study of a single group was carried out, which was approved by the Bioethics and Biosafety Committee of the School of Higher Studies Zaragoza, UNAM, with the number of agreement 23/02-SO/2.4.2 (ISRCTN43215432). All procedures were performed according to the Declaration of Helsinki and with the informed consent of all participants. The study was performed in a convenience sample of 12 older adults, with an average age of 71 ± 6 years (10 women and 2 men) with a diagnosis of MetS according to the National Cholesterol Education Program Adult Treatment Panel III (NCEP/ATP III) criteria [20]. We followed some methods standardized by our research group in previous studies [21].

The following clinical parameters and biochemical markers were measured in the study participants at the beginning of the study and after six weeks of intervention: anthropometric measurements, blood pressure, biochemical parameters (glucose, albumin, renal profile, liver profile and lipid profile), glycosylated hemoglobin (HbA1c), concentration of lipoperoxides, total antioxidant status in plasma (TAS) and erythrocyte activity of superoxide dismutase (SOD) and glutathione peroxidase (GPx) enzymes and inflammatory cytokines in serum.

### 2.2. Anthropometric Measurements

Prior application of complete clinical history, physical evaluation and anthropometric measures were taken according to a standardized protocol by trained personnel. The subjects were weighed wearing only a clinical gown after evacuating, on a Torino calibrated scale. For height measurement, patients were placed with their heels together, buttocks, shoulders and head in contact with the stadiometer with eyes facing the front and the Frankfurt plane parallel to the ground. The subjects’ body mass index (BMI) was calculated through the weight ratio between height squared (kg/m^2^). The circumference of the waist was measured at the level of the umbilical scar, using an asbestos tape measure without putting any pressure on the body.

### 2.3. Blood Pressure

The subjects’ blood pressure (BP) was measured by trained personnel using a mercury manometer on both arms under fasting conditions, or at least two hours after breakfast. Osler’s technique was used to identify pseudo hypertension [22].

### 2.4. Blood Sampling and Biochemical Analyses

Blood samples were collected by venipuncture after a 10-h fast and then placed in vacutainer/siliconized test tubes without anticoagulant for biochemical determinations (glucose, albumin, renal profile, liver profile, lipid profile and cytokines) with ethylene diamine tetraacetic acid (EDTA) as anticoagulant for glycosylated hemoglobin and with heparin for the oxidative stress OxS tests. These were fractioned as follows: 600 μL of whole blood for SOD, 100 μL for GPx, heparinized plasma 100 μL for TAS and 1000 μL for lipid peroxidation (LPO) were separated. The techniques for SOD, TAS and GPx were performed at microscale in multiwell plates, which were read on a Multiskan Go from Thermo Scientific (Vantaa, Finland).

Glucose, albumin, renal profile, liver profile and lipid profile were determined using colorimetric techniques with an automated Selectra Junior clinical chemistry analyzer (Vital Scientific, Dieren, Netherland). For all determinations the intraassay and interassay variation coefficients were less than 5%. An immunoturbidimetric assay was used for the measurement of glycosylated hemoglobin and serum C reactive protein (CRP) in the same chemistry analyzer.

### 2.5. Plasma Thiobarbituric Acid Reactive Substances (TBARS)

The TBARS assay was performed as described by Jentzsch et al. [23]. This test is based on the generation of a pink compound with absorption at 535 nm. The reaction occurs between a molecule of malondialdehyde with two molecules of thiobarbituric acid (TBA) in an acid medium. The possible amplification of the peroxidation during the test is avoided by the addition of the antioxidant). The quantification was done using a calibration curve.

### 2.6. Plasma Total Antioxidant Status

Plasma total antioxidant status quantification was done using 2,2′-azino-bis (3-ethylbenzthiazoline-6-sulfonic acid) (ABTS) (Randox Laboratories Ltd., County Antrim, United Kingdom), which was incubated with a peroxidase to generate the blue-colored radical cation ABTS^+^•. The antioxidants present in the plasma sample cause suppression of this color to a degree proportional to the concentration. The kinetics reaction was measured at 600 nm.

### 2.7. Red Blood Cell Superoxide Dismutase

Superoxide radicals were generated from xanthine and xanthine oxidase. The formed superoxide radical reacts with 2-(4-iodophenyl)-3-(4-nitrophenol)-5-phenyltetrazolium chloride and forms a red color, measured at 505 nm. The enzyme present in the sample causes the inhibition of this reaction, so its activity is proportional to the degree of inhibition. It was measured with a commercial kit from Randox Laboratories Ltd., (County Antrim, UK).

### 2.8. Red Blood Cell Glutathione Peroxidase

The oxidation of glutathione by cumene hydroperoxide is catalyzed by glutathione peroxidase in the presence of glutathione reductase and NADPH. Oxidized glutathione is converted into the reduced form with subsequent oxidation of NADPH to NADP^+^, this decrease in absorbance was measured at 340 nm (Randox Laboratories Ltd., County Antrim, UK).

The SOD/GPx ratio and the antioxidant gap (GAP) were calculated. The GAP was calculated using the equation [24]:AOGAP = (TAS − [(albumin (mmol) × 0.69) + uric acid (mmol)].

### 2.9. Oxidative Stress Score

We defined the cut-off values of each parameter based on the 90th percentile of healthy young subjects: lipid peroxidation (LPO) ≥ 0.340 mmol/L, superoxide dismutase (SOD) ≤ 170 IU/mL, glutathione peroxidase (GPx) ≤ 5500 IU/L, total antioxidant status (TAS) ≤ 0.9 mmol/L, SOD to GPx ratio (SOD/GPx) ≥ 0.023 and antioxidant gap (AOGAP) ≤ 190 mmol/L. A score of 1 was assigned to the values above or under the cut-off and a stress score (OxS) was generated ranging from 1 to 6, representing the severity of the modifications of the included biomarkers [25].

### 2.10. Inflammatory Cytokines and C-Reactive Protein (CRP)

Aliquots of serum samples were assayed by flow cytometry (CBA Kit, Human Inflammatory Cytokine, BD, San Diego, CA, USA) to determine the levels of interleukin (IL), IL1-β, IL-6, IL-8, IL-10 and tumor necrosis factor-alpha (TNF-α). For the measurement of CRP, particles coated with anti-human CRP antibodies were used, which were agglutinated by CRP molecules present in the serum samples analyzed. Since the agglutination causes changes in the absorbance proportionally to the concentration of CRP and after comparison with a calibrator, it was possible to determine the exact concentration of the protein. This test was carried out on the Selectra Junior automated equipment (Vital Scientific, Dieren, Netherland) under a turbidimetric principle, using a commercial kit from Spinreact (CRP Turbi 1107101L; Girona, Spain).

### 2.11. Treatment

The capsules of *S. edule* were formulated and elaborated in the pharmaceutical plant of the FES Zaragoza with biological material (*Sechium edule* var. *nigrum spinosum*) donated by the Interdisciplinary Group for the study of *Sechium edule* de México S.A. (GISEM). The intervention consisted of consuming three capsules of 500 mg of *S. edule* (one before each meal) for six weeks. The capsules were made following good manufacturing practices prior to dust sanitization, which was obtained by grinding the slices of the dried *S. edule* fruit.The content of secondary metabolites was analyzed by high-performance liquid chromatography (HPLC) following the protocol described in Salazar-Aguilar et al. (2017) [26]. The capsules (500 mg) contained cucurbitacins B, D and E (4.91, 0.58 and 169.39 µg, respectively), phenolic acids such as gallic, syringic, vanillinic, caffeic, pherulic and p-coumaric acids (0.36, 7.68, 15.65, 13.87, 11.81 and 4.19 µg, respectively), and flavonoids asrutin, phloridzin, myricetin, quercetin, naringenin and galangin (8.15, 17.0, 5.26, 0.77, 157.55 and 9.57 µg, respectively).

## 3. Statistical Analysis

Averages and standard deviation were calculated; the data were compared by means of the Wilcoxon test using the statistical program IBM SPSS V 20 (Armonk, NY, US.).

## 4. Results

Table 1 shows the clinical and anthropometric parameters pre- and post-intervention; no significant differences were found.

With regard to biochemical markers, a statistically significant decrease in the concentrations of uric acid, creatinine as well as the liver enzymes alanine amino transferase (ALT) and aspartate amino transferase (AST) was observed after treatment (Table 2).

Likewise, in the markers of oxidative stress, a statistically significant decrease in the concentration of lipoperoxides was observed (baseline, 0.289 ± 0.04 vs. post-treatment, 0.234 ± 0.06 μmol/L, *p* < 0.05), with a significant increase of total antioxidants (baseline, 0.97 ± 0.18 vs. post-treatment, 1.2 ± 0.12 mmol/L, *p* < 0.05). In this sense, the oxidative stress index showed a statistically significant decrease (baseline, 1.7 ± 0.78 vs. post-treatment, 0.75 ± 0.87, *p* < 0.05) (Table 3).

Regarding inflammation markers, a statistically significant decrease in TNF-α concentration was observed after treatment (baseline, 5.3 ± 1.4 vs. post-treatment, 3.5 ± 1.3, *p* < 0.05) (Table 4).

## 5. Discussion

A global increase in the population of older adults has been observed, and it is projected to continue in the following years due to a high prevalence of chronic non-communicable diseases (CNCDs), including MetS. In this sense, it is necessary to propose affordable and safe therapeutic alternatives to prevent and control diseases related to aging whose physiopathology is linked to OxS and chronic inflammation [27]. In this regard, in vitro and in vivo studies have reported that the edible fruit of *S. edule* has a wide variety of compounds with antioxidant, anti-inflammatory, hypoglycemic, hypertensive as well as lipogenesis inhibitor effects [18]. As such, it represents an alternative for the complementary treatment of MetS.

MetS has become one of the main public health problems of the 21st century. MetS occurs with high frequency among older adults and is associated with other diseases, mainly with metabolic problems such as type 2 diabetes and cardiovascular diseases [28,29,30].

In this study a decrease in the serum concentration of glucose and triglycerides was observed. Although this difference was not statistically significant, the tendency toward decrease coincides with that reported by other authors. This can be explained given the pharmacological mechanisms of the extracts of *S. edule*.

On the other hand, *S. edule* contains a high quantity of flavonoids, among which are quercetin and epicatechin, for which a hypoglycemic effect secondary to the increase of insulin release via the modification of calcium metabolism in Langerhans cells has been reported [31,32]. It has also been shown that *S. edule* extract decreases lipid synthesis through the signaling pathway of AMPK which inhibits the expression of lipogenic enzymes which stimulates the expression of PPARα and CPT I, critical in the regulation of the hepatic metabolism of lipids—an effect that is attributed to the activity of the polyphenols present in the extract [33,34].

With respect to markers of renal function, a significant decrease in creatinine and uric acid concentrations was observed in the present study, which is consistent with what was reported by Firdous et al. (2013) who found that the administration of *S. edule* extract in rats with induced kidney damage generated a significant decrease in creatinine, urea and uric acid, accompanied by an improvement in renal histology of both tubules and glomeruli [35]. These changes allow us to suppose that the assets present in *S. edule* modify renal structure and function, hence a greater clarification of the parameters favors the decrease of serum levels, which coincides with our results. This finding is relevant considering that in recent studies it has been shown that the metabolic alterations present in MetS are associated with renal damage in the microstructure by various mechanisms, including OxS. In this sense, it has been reported that the administration of flavonoids of a natural origin has mitigated OxS, so we can suggest that the active substances present in *S. edule* are acting synergistically, and that they have a renal-protective effect of clinical importance and could be an alternative in the prevention of kidney damage [36,37,38].

With regard to the concentration of the liver enzymes AST and ALT, a statistically significant decrease was observed. These results suggest a hepatoprotective effect, which has also been reported by Firdous (2012) in both mice and rats with induced liver damage [39]. These enzymes are considered a marker of liver function since their increase in serum is due to their escape from the hepatocyte into the circulation by an increase in membrane permeability. In this sense, the hepatoprotective effect and probably the improvement observed at the renal level could be associated with the flavones present in *S. edule*, since the active ingredients such as apigenin, quercetin and naringenin have been identified as possible therapeutic agents against tissue damage by various mechanisms, among which is the regulation of the synthesis of phospholipids in the membrane, the prevention of oxidative damage and the diminution of proinflammatory cytokine release, as has been reported by other researchers [40,41,42,43,44].

Regarding OxS markers, in our study we observed a significant decrease in the concentration of lipoperoxides and the overall OxS-score accompanied by a significant increase in total antioxidant capacity and AOGAP. As previously noted, it has been consistently reported that different extracts of *S. edule* contain a wide variety of bioactive compounds, among which are polyphenols, such as gallic, chlorogenic, vanillinic, caffeic and coumaric acid, flavonoids such as phloridzin, naringenin, floretin and apigenin. These molecules have also been isolated from other fruits, and have been investigated, finding that they act as potent antioxidants and anti-inflammatory agents [45].

Regarding flavonoids (specifically the effect of naringenin), at least three mechanisms of antioxidant action have been described. At the extracellular level, naringenin directly interacts with free radicals through its OH groups by transferring hydrogen to the free radical to stabilize the molecule, while the 5,7-dihydroxy group in ring A of naringenin increases the stability of the molecule via electronic resonance. At the cellular level, naringenin accumulates in the middle of the lipid bilayer and interacts with the nonpolar lipid tail due to its lipophilic properties, which favor the maintenance of membrane rigidity and reduce lipid peroxidation. At the nuclear level, naringenin decreases the expression of the microRNA miR-17-3p that functions to inhibit the expression of the SOD, GPx and CAT genes, which leads to a decrease in OxS. Although the mechanisms of all isolated *S. edule* actives have not been fully described, it has been pointed out that the antioxidant effects show a significant correlation with the content of phenols, and it is very likely that the different compounds act synergistically, such that the antioxidant effect observed in our results was achieved [46,47,48,49,50,51,52,53].

Finally, the anti-inflammatory effect has also been reported. The mechanism by which it is explained involves the ability of the flavonoids to block key molecules in inflammation and prothrombotic processes, such as nuclear factor kappa B (NF-κB). Likewise, it has been reported that the PPARa factor, which is stimulated by the polyphenols present in *S. edule*, interferes with the signaling of pro-inflammatory transcription factors, including the signal transducer and activator of transcription (Stat), the activator protein-1 (AP-1) and NF-κB, from which derives the proposal to investigate agonist molecules of this receptor in order to find substances with therapeutic potential to treat inflammatory liver diseases. According to our findings in this exploratory study, the synergistic action of the active substances present in *S. edule* has a metabolic, antioxidant and anti-inflammatory effect in older adults with MetS, which makes it a suitable safe option for older adults, whose treatment must be comprehensive and must assess the risk benefit given the characteristics of this population group [54,55,56,57,58,59].

Among the most important limitations of the study, we can point out that the sample size was not representative and a placebo group was not included, although considering that it is an exploratory study, the findings are relevant and may be useful as a background to propose a randomized clinical trial.

## 6. Conclusions

Our findings suggest that the consumption of the dry fruit of *S. edule* has an antioxidant and anti-inflammatory effect in older adults with MetS, which justifies a continued investigation by increasing the sample size and the duration of the intervention.

## Figures and Tables

**Table 1 antioxidants-08-00146-t001:** Anthropometric characteristics before and after treatment.

Parameter	Baseline	Post-Treatment
BMI	29.3 ± 4.4	29.4 ± 4.4
SBP (mmHg)	127 ± 12	127 ± 16
DBP (mmHg)	82 ± 7	81 ± 9
Circumference of the waist (cm)	99.5 ± 9	99 ± 7

Data are expressed as means ± standard deviation. Wilcoxon test, significance level 95%, *p* > 0.05. BMI: body mass index; SBP: systolic blood pressure; DBP: diastolic blood pressure.

**Table 2 antioxidants-08-00146-t002:** Markers pre- and post-treatment.

Parameter	Baseline	Post-Treatment
Glucose (mg/dL)	107 ± 38	95 ± 5
Total Cholesterol (mg/dL)	204 ± 34	200 ± 28
Triglycerides (mg/dL)	170 ± 76	153 ± 66
HDL-C (mg/dL)	49 ± 7	50 ± 9
Uricacid (mg/dL)	5.1 ± 1.2	4.2 ± 1.2 *
Urea (mg/dL)	38 ± 22	35 ± 18
Creatinine (mg/dL)	1.1 ± 0.26	0.85 ± 0.2 *
Albumin (mg/dL)	4.2 ± 0.2	4.2 ± 0.15
AST (U/L)	30.9 ± 11	24.8 ± 10 *
ALT (U/L)	36 ± 18	28 ± 12 *
Total Bilirubin (mg/dL)	0.61 ± 0.27	0.62 ± 0.22
Direct Bilirubin (mg/dL)	0.22 ± 0.098	0.26 ± 0.08
HbA1c (%)	5.9 ± 2.7	6.0 ± 2.7

Data are expressed as means ± standard deviation. Wilcoxon test, significance level 95%, * *p* < 0.05. HDL-C: high-density lipoprotein cholesterol; AST: aspartate amino transferase; ALT: alanine amino transferase; HbA1c: glycosylated hemoglobin.

**Table 3 antioxidants-08-00146-t003:** Markers of oxidative stress before and after treatment.

Parameter	Baseline	Post-Treatment
Lipoperoxides (µmol/L)	0.289 ± 0.04	0.234 ± 0.06 *
SOD (U/mL)	190 ± 3.4	190 ± 8
GPx (U/L)	7542 ± 2651	8113 ± 3477
TAS (mmol/L)	0.97 ± 0.18	1.2 ± 0.12 *
SOD/GPx	0.27 ± 0.013	0.28 ± 0.011
AOGAP (µmol/L)	188 ± 258	497 ± 83 *
OxS-Score	1.7 ± 0.78	0.75 ± 0.87 *

Data are expressed as means ± standard deviation. Wilcoxon test, significance level 95%, * *p* < 0.05. SOD: superoxide dismutase; GPx: glutathione peroxidases; TAS: total antioxidant status, SOD/GPx: SOD/GPx ratio, AOGAP: antioxidant gap; OxS-Score: oxidative stress score.

**Table 4 antioxidants-08-00146-t004:** Markers of inflammation before and after treatment.

Parameter	Baseline	Post-Treatment
IL-12p70 (pg/dL)	1.8 ± 0.9	1.2 ± 1.1
TNF-α (pg/dL)	5.3 ± 1.4	3.5 ± 1.3 *
IL-10 (pg/dL)	1.7 ± 0.65	1.3 ± 0.86
IL-6 (pg/dL)	4.2 ± 0.8	3.3 ± 1.6
IL-1β (pg/dL)	8.5 ± 1.4	9.5 ± 2.7
IL-8 (pg/dL)	11.5 ± 3.9	11.8 ± 3.6
CRP (mg/dL)	0.36 ± 0.31	0.30 ± 0.32

Data are expressed as means ± standard deviation. Wilcoxon test, significance level 95%, * *p* < 0.05.IL: interleukin; CRP: C-reactive protein.

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
