# Peer review of "Effect of Sechium edule var. nigrum spinosum (Chayote) on Oxidative Stress and Pro-Inflammatory Markers in Older Adults with Metabolic Syndrome: An Exploratory Study"

_antioxidants, 2019, doi:10.3390/antiox8050146_

Reviewer 1 Report

This study investigate the antioxidant and anti-inflammatory  properties of  Sechium edule (chayote) in 12 older adults, with an average age of 71±6 years with a diagnosis of MetS.  Despite the main limitation of the study is the sample size, the results demonstrated a synergistic action of the active substances present in S. edulen in older adults with MetS. However some minor issues need to be addressed:

·   Please improve the discussion with recent publications:

Line 207: Please enrich the following sentence “With respect to markers of renal function, a significant decrease in creatinine and uric acid concentrations was observed in the present study, which is consistent with that reported by Firdous et al. (2013) who found that the administration of S. edule extract in rats with induced kidney damage generated a significant decrease in creatinine, urea and uric acid, accompanied by an improvement in renal histology of both tubules and glomeruli”  with  recent experimental evidences investigating both the  detrimental role of some clusters of metabolic syndrome such as  oxidative stress, hypertension in inducing organ damage (PMID:28495910;  PMID:29162432; PMID:30854019) and the  protective role of antioxidants in renal damage induced by metabolic syndrome. (PMID:30884780) 

Line214: Please improve the discussion regarding the hepatoprotective effect of Sechium edule with a recent experimental evidence (PMID: 30388763) investigating the anti-inflammatory activity of other antioxidants in NASH liver.

For future studies I suggest you to evaluate if the effects of Sechium edule on oxidative stress biomarkers are influenced by gender, lifestyle and  smoking habits, use of statins or other drugs and improve clinical information

Author Response

Comment

This study investigate the antioxidant and anti-inflammatory  properties of  Sechium edule (chayote) in 12 older adults, with an average age of 71±6 years with a diagnosis of MetS.  Despite the main limitation of the study is the sample size, the results demonstrated a synergistic action of the active substances present in S. edulen in older adults with MetS. However some minor issues need to be addressed:

·    Please improve the discussion with recent publications:

Line 207: Please enrich the following sentence “With respect to markers of renal function, a significant decrease in creatinine and uric acid concentrations was observed in the present study, which is consistent with that reported by Firdous et al. (2013) who found that the administration of S. edule extract in rats with induced kidney damage generated a significant decrease in creatinine, urea and uric acid, accompanied by an improvement in renal histology of both tubules and glomeruli”  with  recent experimental evidences investigating both the  detrimental role of some clusters of metabolic syndrome such as  oxidative stress, hypertension in inducing organ damage (PMID:28495910;  PMID:29162432; PMID:30854019) and the  protective role of antioxidants in renal damage induced by metabolic syndrome. (PMID:30884780)

Response

We analyze the suggested articles and include the requested information.

Line214: Please improve the discussion regarding the hepatoprotective effect of Sechium edule with a recent experimental evidence (PMID: 30388763) investigating the anti-inflammatory activity of other antioxidants in NASH liver.

Response

We analyze the suggested article and include the requested information.

Comment

For future studies I suggest you to evaluate if the effects of Sechium edule on oxidative stress biomarkers are influenced by gender, lifestyle and  smoking habits, use of statins or other drugs and improve clinical information

Response

Thank you very much for the suggestion, this will be considered in the clinical trial that we will carry out in the coming months.

Reviewer 2 Report

The administration of "chayote" seems to be beneficial to reduce the oxidative stress and inflamation associated to metabolic síndrome. Although, the obtained results have a potential interest, I do not feel that the manuscript deserves publication in the journal. I recommend to consider a new intervention design, including diet and exercise in order to get a weight reduction and see the long-term effect of the plant under more clinical conditions. In thse present form, circulating lipids remain still very high and the no reduction of these paremeters is highly related with cardivascular problems probably due to a disruption of oxidative equilibrium. Otherwise said, the study needs a more realistic intervention closest to the clinical practice in order to undrestand the clinical potential of "chayote".

The authors need to put the results in a real context. The reality is that the patients do not improve some parameters and it is important to discuss why and which are the complementary strategies to introduce this treatment anti-obesity.

Author Response

Comment

The administration of "chayote" seems to be beneficial to reduce the oxidative stress and inflamation associated to metabolic síndrome. Although, the obtained results have a potential interest, I do not feel that the manuscript deserves publication in the journal. I recommend to consider a new intervention design, including diet and exercise in order to get a weight reduction and see the long-term effect of the plant under more clinical conditions. In thse present form, circulating lipids remain still very high and the no reduction of these paremeters is highly related with cardivascular problems probably due to a disruption of oxidative equilibrium. Otherwise said, the study needs a more realistic intervention closest to the clinical practice in order to undrestand the clinical potential of "chayote".

The authors need to put the results in a real context. The reality is that the patients do not improve some parameters and it is important to discuss why and which are the complementary strategies to introduce this treatment anti-obesity.

Response

It should be considered that it is a "pilot study" whose findings will be very useful for a larger investigation.

Thank you very much for your comments and suggestions, which will be very useful for the clinical trial that we will carry out in the coming months.

Reviewer 3 Report

This study shows the effect of the vegetable, chayote, on preventing increase in oxidation and inflammatory markers in some elder people with a diagnosis of metabolic syndrome. The authors have assessed the levels of oxidative markers such as a lipoperoxide, and inflammatory cytokines such as TNF and IL-6 in this study. This study is interesting. However, the reviewer considered that there were some weak points so that this article should be revised a bit before publication.

1. The reviewer wondered how much flavonoid the chayote used in this study contained. The authors should have shown which flavonoid is contained as well as its level.

2. In L 193-223, those parts seem irrelevant. Because this study focuses on the anti-oxidative and anti-inflammatory effect of chayote, not its influence on glucose metabolism and diabetes.

3. The authors should show how the patients took in chayote powder(?) in this study, even though they had described it in their previous paper.

4. In Table 4, is ‘PCR’ correct?  Is rather 'CPR' right?

Author Response

This study shows the effect of the vegetable, chayote, on preventing increase in oxidation and inflammatory markers in some elder people with a diagnosis of metabolic syndrome. The authors have assessed the levels of oxidative markers such as a lipoperoxide, and inflammatory cytokines such as TNF and IL-6 in this study. This study is interesting. However, the reviewer considered that there were some weak points so that this article should be revised a bit before publication.

Comment

1. The reviewer wondered how much flavonoid the chayote used in this study contained. The authors should have shown which flavonoid is contained as well as its level.

Response

We have included the data of the flavonoid content of this variety of chayote

2. In L 193-223, those parts seem irrelevant. Because this study focuses on the anti-oxidative and anti-inflammatory effect of chayote, not its influence on glucose metabolism and diabetes.

Response

We included the information regarding the hypoglycaemic effect of chayote feeding considering what was reported in other studies, so that it could potentially be found in a larger sample.

3. The authors should show how the patients took in chayote powder(?) in this study, even though they had described it in their previous paper.

Response

This was included

4. In Table 4, is ‘PCR’ correct?  Is rather 'CPR' right?

Response

This was corrected

Round  2

Reviewer 2 Report

The manuscript has improved significantly and it can be considered for publication in Antioxidants

Reviewer 3 Report

The reviewer appreciates the response and effort which has been done by the authors to enhance the quality of the work. After the check and correction of misspelling is done, the manuscript would be acceptable.